# Measuring for primary prevention: An online survey of local community perspectives on family and domestic violence in regional Australia

**John A. Woods**[1]*, **Andrew C. Ward**[2], **Heath S. Greville**[1], **Monica C. Moran**[1], **Barbara Nattabi**[3], **Karen E. Martin**[4], **Sandra C. Thompson**[1]

**1** Western Australian Centre for Rural Health, School of Allied Health, The University of Western Australia, Geraldton, Western Australia, Australia, **2** The Social Research Centre, Melbourne, Victoria, Australia, **3** School of Population and Global Health, The University of Western Australia, Perth, Western Australia, Australia, **4** School of Education, The University of Tasmania, Hobart, Tasmania, Australia

* john.woods@uwa.edu.au

**Data Availability Statement:** Data cannot be shared publicly because of confidentiality restrictions. All data used for the analysis

## Abstract

### Background

Family and domestic violence, encompassing diverse behaviours including physical, sexual, emotional and financial abuse, is endemic worldwide and has multiple adverse health and social consequences. Principal drivers include traditional gender values that disempower women. Changing these is a key prevention strategy. In Australia, high-quality national surveys provide data on public perspectives concerning family and domestic violence but may not capture community-level diversity. As part of a project for primary prevention family and domestic violence in outer regional Australia, our aims were to develop and administer a questionnaire-based survey suitable for the local community encompassing knowledge about, attitudes towards, and personal experiences of family and domestic violence, to describe and to investigate the theoretical (factor) structure and local socio-demographic predictors of responses, and to determine the extent to which the survey findings are locally distinctive.

### Methods

The online community survey for local residents (≥15 years), comprised items on respondents' sociodemographic characteristics plus questions abridged from pre-existing national instruments on knowledge about, attitudes towards, and personal experiences of family and domestic violence. Responses were rake-weighted to correct census-ascertained sample imbalance and investigated using exploratory factor analysis, with sociodemographic predictors determined using multiple linear regression and dominance analysis.

presented are accessible only to investigators involved in the study who have been given appropriate clearance by the Human Research Ethics Committee of the University of Western Australia, which approved the research project (Research approval #: RA/4/20/4860, 13/10/2019). Access to the data for researchers who meet the criteria for access to confidential data can be sought through an application process by contacting the Human Ethics Office at the University of Western Australia on (+61) 8 6488 3703 or by emailing to humanethics@uwa.edu.au.

**Funding:** The 'Conversations for Change: Accelerating efforts to prevent family violence' project was supported by Healthway grant 31994 (SCT, MCM, BN, KEM) (https://www.healthway.wa. gov.au/). Preliminary work that assisted with the development of LCAEVS was supported by a small grant from Desert Blue Connect (SCT, MCM, BN, KEM) (https://desertblueconnect.org.au/) that was matched by philanthropic funds administered by the School of Population and Global Health at The University of Western Australia (SCT, MCM, BN, KEM) (https://www.uwa.edu.au/schools/ population-global-health). The funders had no role in study design, data collection and analysis, decision to publish, or preparation of the manuscript.

**Competing interests:** The authors have declared that no competing interests exist.

## Results

Among 914 respondents, males (27.0%), those from age-group extremes, and less-educated persons were underrepresented. Familiarity with diverse family and domestic violence behaviours was high among all subgroups. Poorer knowledge of the FDV behaviour continuum and attitudes supporting traditional gender roles and FDV were disproportionately evident among males, older respondents and those with lower education levels. Both the factor structure of extracted composite measures reflecting community perspectives and sociodemographic predictors of responses generally aligned with patterns evident in national data.

## Conclusions

Local reinforcement of existing nationwide findings on community understanding of and attitudes towards family and domestic violence provides salience for targeted interventions.

## Introduction

Family and domestic violence (FDV) has long been recognised as a major global public health problem [1]. FDV encompasses behaviours comprising 'acts of physical aggression, sexual coercion, psychological abuse and controlling behaviours' between intimate partners [2] (p11) as well as abuse (physical, sexual, neglect, verbal or emotional) or neglect of a child or elder [3]. Fundamental to FDV is coercive controlling behaviour, which is consistently linked with risk to family members of physical violence, including homicide [4].

The frequency of FDV is difficult to determine accurately because of inconsistencies in definition and measurement [5] as well as constraints on recognition and barriers to disclosure and help-seeking behaviours among victims [6], with under-reporting likely to be substantial. However, FDV is believed to be extremely common in Australia. In the most recent national survey data, one in six women and one in sixteen men reported subjection to physical or sexual violence by a current or previous cohabiting partner [7]. Based on older estimates from the Australian Component of the International Violence Against Women Survey (2004), over one-third of women who had a current or former partner reported at least one lifetime experience of intimate partner violence [8].

A multilevel framework is required for conceptualising and addressing the complex causes of FDV, which encompass collective norms, practices and structures (Counting on Change p. 44) as well as the psychological traits of individual perpetrators [1, 9]. A public health approach using a social ecology framework can aid in understanding the complex relationships between individuals and their environments, and thereby identify appropriate targets for interventions. This framework identifies four levels within which drivers of violence operate: societal, community, relationship, and individual [10]. Traditional gender values that emphasise rigid masculine stereotypes and disempower women are strongly implicated as underlying causes of FDV [11, 12].

Given the diversity of socio-cultural and economic influences on violence, there are local variations within countries as well as global differences [13], necessitating local-level investigation and intervention. In Australia and similar economically developed countries, non-urban communities may be characterised by distinctive demographic profiles as well as local cultural discourses and values that influence the perpetration of and responses to violence, including conservative and religious family values, self-reliance, and domestic privacy [14, 15]. In

addition, women's experiences of violence are strongly associated with poor social support and with the financial stress arising from economic disadvantage and unemployment [16], both of which are commonly accentuated in rural areas. Finally, paucity of professional and informal support services and heightened threats to anonymity act as barriers to disclosure and help-seeking in non-urban communities [17, 18].

The centrality of deep-rooted societal factors as primary causes of FDV [19–21] under-scores the importance of primary prevention for reducing its occurrence and impact. For optimal effectiveness, preventive strategies must be directed concurrently against these key drivers of violence that shape societal structures, community perspectives, and behaviours [22]. In turn, optimally focussed targeting of such interventions requires investigation using validated survey instruments of community perspectives in relation to these drivers impacting across the social ecology [23] and towards violent behaviours [24]. In Australia, periodic nationwide surveys focussing on attitudes towards gendered drivers are undertaken using high-quality validated instruments. The *National Community Attitudes to Violence Against Women Survey* (NCAS) [25, 26] is a telephone questionnaire developed by Australia's National Research Organisation for Women's Safety Limited (ANROWS) and conducted periodically on large, representative nationwide samples. Personal experiences of FDV are investigated with the *Personal Safety Survey* (PSS), which is conducted periodically by the Australian Bureau of Statistics with personal face-to-face interviews [7, 27]. However, these surveys may not provide data with sufficient granularity to reveal patterns at the local level across socio-demographically diverse communities nationwide.

Local data can inform the targeting and features of primary preventive interventions and also provide a baseline from which to evaluate their effectiveness. The current study presents the results of a community survey on local perspectives towards FDV in a non-urban Australian setting. The *Local Community Attitudes and Exposures to Violence Survey* (LCAEVS) instrument was designed as a tool to guide the targeting of a program for the primary prevention of FDV in the inner regional Local Government Area of Greater Geraldton, Western Australia (population ~39,000). Our aims were: (i) to develop and administer a questionnaire-based survey suitable for the local community encompassing knowledge about, attitudes towards, and personal experiences of FDV, (ii) to describe community responses to this questionnaire, and to investigate the theoretical (factor) structure and local socio-demographic predictors of these responses, and (iii) to determine the extent to which the survey findings are locally distinctive.

## Materials and methods

### Procedures: Questionnaire development, sampling and recruitment

LCAEVS was designed to examine community perspectives on core FDV issues, with a view to enabling nationwide comparability, while minimising respondent burden for on-line completion. The content of LCAEVS was adapted principally from NCAS, but drew also from the PSS, and from a community-level survey conducted in Wagga Wagga, New South Wales [28], along with relevant documents published by the World Health Organization [2, 21]. LCAEVS was administered online during the period 20/10/2019–30/11/2019 using the Qualtrics platform (Qualtrics Corporation, Provo, UT, USA), and was hosted on the website of the local university centre at which the researchers were based. Additionally, a small number of surveys was made available in hard copy for persons with difficulty accessing the internet, for example those in prison. The survey was promoted throughout the local community in late 2019 via e-mail lists, Facebook (organisational accounts and paid targeted promotion), at local events, in local newspapers, on local radio, through printed flyers and posters, and through letter

box drops to certain suburbs through Australia Post. In order to encourage survey completion, respondents were given the opportunity to enter a cash prize draw. To ensure confidentiality, identifying details of persons participating in the prize draw were stored in a database separate from the survey response data. Persons eligible to participate in the survey were those who identified as a resident of Greater Geraldton and were aged at least fifteen years.

The research and intervention project of which the survey is a component were approved by the Human Research Ethics Committee of the University of Western Australia (RA/4/20/4860, 13/10/2019). Prior to providing responses to the questionnaire, all respondents provided online written consent, 'agreeing to participate in this study as outlined in the information sheet that is made available . . . by the research team'. The potential for triggering by participation in the survey was highlighted explicitly with a statement at the beginning of the online and print versions of the questionnaire that 'some of the questions relate to different types of violence and may be upsetting to some people'. Additionally, telephone numbers for relevant helplines and web chat services were provided at the beginning of the questionnaire.

## Measures

The questionnaire comprised items on the sociodemographic characteristics of respondents in addition to FDV-related items. Options for gender identity were Male, Female or Other. Age was recorded in years and for analysis was collapsed into decadal age-group categories, except for respondents aged >65, who constituted a single category. In relation to reported Aboriginal and/or Torres Strait Islander identification, respondents were categorised as Indigenous (if identifying as either or both Aboriginal and/or Torres Strait Islander) or non-Indigenous. Respondents' country of birth was categorised as Australia, non-Australia Anglosphere (i.e., NZ, UK, Ireland/Eire, South Africa, Canada, or USA) or Other. Respondents' highest reported education level attained was recorded as one of six categories (collapsed into four for analysis) and their reported current annual income was divided into five categories (Table 1).

FDV-related items (Table 2) comprised (i) eighteen questions about recognition of different specific behaviours being FDV, (ii) twenty-one questions about attitudes towards violence, and (iii) eight questions about personal experiences of being a FDV victim.

## Statistical analysis

All analyses were conducted using Stata version 16.1 (Stata Corporation, College Station, TX, USA) and were restricted to questionnaires in which respondents had completed all items. Predictors of non-completion were investigated using multivariable robust Poisson regression [29]. Preliminary examination of the survey data indicated that the profile of respondents was imbalanced in relation to certain core demographic attributes by comparison with the resident population of the City of Greater Geraldton as ascertained from the most recently reported (2016) national Census data [30]. Given the uncertainties in relation to the selection mechanisms that influenced sampling, we assigned weights to the data in order to (i) reduce bias, and (ii) allow data inferences in relation to the target population. We adopted a 'superpopulation' approach [31] (p115) by means of an iterative proportional fitting ('raking') procedure [32]. Data were weighted in respect to respondents' self-reported gender, highest education level, Indigenous identifier and age group using the ipfweight command in Stata®.

Sociodemographic characteristics of respondents were summarised. All subsequent analyses were performed on the weighted dataset. Responses to individual items were collapsed into binary form according to item type (e.g., Strongly Agree/Agree versus all other responses in the case of items with five-tiered ordered options reflecting degree of agreement). The crude

**Table 1. Characteristics of survey respondents (N = 914) in comparison with greater geraldton population, by indigenous identifier.**

| | | Non-Indigenous | | Indigenous | | Total sample | | Population | Analysis weight |
|---|---|---|---|---|---|---|---|---|---|
| | | n | % | n | % | n | % | % | |
| **Indigenous Identifier** | Indigenous | | | | | 77 | 8.4 | 9.7 | 1.15 |
| | Non-Indigenous | | | | | 837 | 91.6 | 90.3 | 0.99 |
| **Gender*** | Male | 216 | 25.8 | 31 | 40.3 | 247 | 27.0 | 49.6 | 1.84 |
| | Female | 621 | 74.2 | 46 | 59.7 | 667 | 73.0 | 50.4 | 0.69 |
| **Age group** | 15–24 years | 71 | 8.5 | 12 | 15.6 | 83 | 9.1 | 16.2 | 1.78 |
| | 25–34 years | 137 | 16.4 | 21 | 27.3 | 158 | 17.3 | 15.5 | 0.90 |
| | 35–44 years | 187 | 22.3 | 14 | 18.2 | 201 | 22.0 | 16.2 | 0.74 |
| | 45–54 years | 209 | 25.0 | 14 | 18.2 | 223 | 24.4 | 18.3 | 0.75 |
| | 55–64 years | 165 | 19.7 | 14 | 18.2 | 179 | 19.6 | 15.3 | 0.78 |
| | 65+ years | 68 | 8.1 | 2 | 2.6 | 70 | 7.6 | 18.6 | 2.45 |
| **Highest education level attained** | Year 10 or below | 103 | 12.3 | 23 | 29.9 | 126 | 13.8 | 29.4 | 2.13 |
| | Year 12 | 106 | 12.7 | 20 | 26.0 | 126 | 13.8 | 25.1 | 1.82 |
| | Trade/Apprenticeship/TAFE | 280 | 33.5 | 23 | 29.9 | 303 | 33.2 | 33.0 | 0.99 |
| | University | 348 | 41.6 | 11 | 14.3 | 359 | 39.3 | 12.5 | 0.32 |
| **Country of birth** | Australia | 664 | 79.3 | 77 | 100.0 | 741 | 81.1 | N/A | |
| | Non-Australia Anglosphere* | 127 | 15.2 | 0 | 0.0 | 127 | 13.9 | N/A | |
| | Other | 46 | 5.5 | 0 | 0.0 | 46 | 50.3 | N/A | |
| **Annual income** | Under $25,000 | 100 | 12.0 | 22 | 28.6 | 122 | 13.4 | N/A | |
| | $25,000 to $50,000 | 131 | 15.7 | 17 | 22.1 | 148 | 16.2 | N/A | |
| | $51,000 to $100,000 | 276 | 33.0 | 32 | 41.6 | 308 | 33.7 | N/A | |
| | $101,000 to $150,000 | 194 | 23.2 | 3 | 3.9 | 197 | 21.6 | N/A | |
| | Over $150,000 | 136 | 16.3 | 3 | 3.9 | 139 | 15.2 | N/A | |

*Zero respondents entered the 'Other' option provided for gender

N/A not available

proportions with confidence intervals were then estimated using Fisher's exact test, converted to percentages, and displayed graphically.

To investigate constructs captured by the survey, the factor structure of items on family and domestic violence was examined by exploratory factor analysis (EFA) conducted with oblique Promax rotation on polychoric correlation matrices [33] derived from the weighted response data. A positive definite correlation matrix could not be produced with all relevant items analysed simultaneously, so the included items were grouped on *a priori* grounds by domain (knowledge of FDV types and causes, attitudes, personal experiences) and EFA was then performed on each subset. Data were tested for sampling adequacy using the Kaiser-Meyer-Olkin measure [34]. The number of factors to be retained was determined by means of Horn parallel analysis and confirmed by examination of the scree plot [34]. Salient factor loadings were defined using the following rule: primary factor loading $\geq$ .40, alternative factor loading $\leq$ .30 and primary-alternative factor loading difference $\geq$ .20 [35]. Factors comprising at least three measured variables were retained [34], and items that did not load unambiguously onto any factor were investigated separately in multivariable analysis models (below). For sensitivity analysis, EFAs on the original unweighted data were also performed.

The independent influence of sociodemographic determinants on responses to these composite outcomes was investigated using multiple linear regression models with robust variance to account for heteroskedasticity, after investigation of data to confirm normality of residuals and for exclusion of multicollinearity using the variance inflation factor. The models were

**Table 2. Exploratory factor analyses[a].**

| Survey item | Factor[b] loading | | | | | |
|---|---|---|---|---|---|---|
| | KT | KC | Att1 | Att2 | Att3 | Exp |
| **KNOWLEDGE–TYPES OF FDV DOMAIN** | | | | | | |
| **Do you consider the following behaviours to be family and domestic violence? When a person . . .** | | | | | | |
| Slaps or pushes their partner to cause harm or fear | **0.928** | | | | | |
| Tries to scare or control their partner by threatening to hurt other family members | **0.973** | | | | | |
| Throws or smashes objects near their partner to frighten or threaten them | **0.946** | | | | | |
| Repeatedly criticises their partner to make them feel bad or useless | **0.932** | | | | | |
| Controls their partner's social life by preventing them from seeing family and friends | **0.962** | | | | | |
| Tries to control their partner by denying them money | **0.923** | | | | | |
| Repeatedly keeps track of their partner or former partner's location, calls or activities | **0.919** | | | | | |
| Repeatedly follows or watches a partner or former partner | **0.946** | | | | | |
| Repeatedly sends partner or former partner unwanted phone calls, emails, text messages and the like | **0.951** | | | | | |
| Forces their partner to have sex | **0.946** | | | | | |
| **KNOWLEDGE–CAUSES OF FDV DOMAIN** | | | | | | |
| **How much do you agree or disagree that the following cause family and domestic violence?** | | | | | | |
| Alcohol | | **0.937** | | | | |
| Lack of employment | | **0.676** | | | | |
| Use of illegal drugs | | **0.859** | | | | |
| Men's loss of role and status | | 0.390 | | | | |
| Men wanting to control women | | 0.221 | | | | |
| Violence in the community | | 0.291 | | | | |
| Violence in the media (movies, video games etc.) | | -0.084 | | | | |
| Pornography | | -0.126 | | | | |
| **ATTITUDES DOMAIN** | | | | | | |
| **How much do you agree or disagree with the following statements?** | | | | | | |
| I would tell a male friend when his behaviour is sexist | | | 0.131 | -0.081 | 0.065 | |
| If I saw a male friend insulting or verbally abusing a woman he is in a relationship with I would do or say something to show I didn't approve | | | -0.114 | 0.173 | 0.087 | |
| If a person hits you, you should hit them back | | | -0.156 | 0.102 | **0.703** | |
| It is okay to hit children if they have done something wrong | | | 0.065 | -0.045 | **0.551** | |
| If people threaten my family / friends they deserve to get hurt | | | 0.106 | -0.073 | **0.764** | |
| Family and domestic violence is a private matter to be handled in the family | | | **0.783** | 0.054 | -0.189 | |
| Family and domestic violence can be excused if afterwards the person genuinely regrets what they have done | | | **0.878** | -0.018 | -0.013 | |
| If a woman reports abuse by her partner to outsiders it is shameful for her family | | | **0.650** | 0.060 | -0.076 | |
| Many women tend to exaggerate the problem of male violence | | | **0.536** | 0.165 | 0.115 | |
| If a woman is raped while she is drunk or affected by drugs she is at least partly responsible | | | **0.603** | -0.060 | 0.160 | |
| A woman who does not leave an abusive partner is partly responsible for the abuse continuing | | | **0.443** | 0.112 | 0.244 | |
| Women often say 'no' when they mean 'yes' | | | **0.555** | 0.060 | 0.186 | |
| If a woman wears revealing clothing, she is at least partly responsible for rape | | | **0.576** | 0.099 | 0.163 | |
| I'm more likely to listen to a man's opinion than a woman's | | | 0.336 | 0.326 | -0.116 | |
| It's ok for men to whistle at women while they are walking down the street. | | | 0.078 | 0.284 | 0.187 | |
| I think there's no harm in men making sexist jokes about women when they are among their male friends | | | -0.075 | **0.467** | 0.279 | |
| Men should take control in relationships and be the head of the household | | | 0.105 | **0.728** | 0.031 | |
| Men make better political leaders than women | | | 0.007 | **0.937** | -0.049 | |
| In the workplace, men generally make more capable bosses than women | | | 0.023 | **0.905** | -0.052 | |
| Women seek to gain power by gaining control over men | | | 0.148 | **0.681** | 0.035 | |
| Violence by a man against his female partner can be excused if the offender is heavily intoxicated by alcohol | | | **0.615** | 0.162 | -0.095 | |

*(Continued)*

**Table 2.** (Continued)

| Survey item | Factor[b] loading | | | | | |
|---|---|---|---|---|---|---|
| | KT | KC | Att1 | Att2 | Att3 | Exp |
| **EXPERIENCE DOMAIN** | | | | | | |
| **When, if ever, was your most recent experience of any of these from someone you were in a relationship with (dating, de-facto, or married)?** | | | | | | |
| Slapped, pushed or hit you | | | | | | **0.727** |
| Forced you to have sex | | | | | | **0.893** |
| Threatened to hurt other family members | | | | | | **0.880** |
| Repeatedly criticised you to make you feel bad or useless | | | | | | **0.774** |
| Threw or smashed an object near you to cause fear | | | | | | **0.820** |
| Prevented you from seeing friends or family | | | | | | **0.838** |
| Tracked your location without your consent or monitored your phone calls and messages | | | | | | **0.775** |
| Repeatedly sent you unwanted phone calls, emails, text messages and the like | | | | | | **0.866** |

[a]Bold font indicates unambiguous loading of item onto a single factor. Factor loadings based on polychoric factor analysis of rake-weighted data, with promax rotation. A positive definite correlation matrix could not be produced from all relevant items analysed simultaneously, so the included items were grouped on a priori grounds by domain and EFA was then performed on each subset.

[b]Factor abbreviations:

KT: Knowledge–Types of FDV

KC: Knowledge–Causes of FDV

Att1: Attitudes 1 Domestic violence minimising

Att2: Attitudes 2 Male chauvinism

Att3: Attitudes 3 General violence

Exp: Experience

designed to replicate partially those used in NCAS to maximise comparability of results with those of the national survey. Finally, for selected regression models, dominance ('relative importance') analysis [36, 37] was used to calculate the relative contributions of each predictor to the variance explained ($R^2$). Concomitantly, investigation of the sociodemographic predictors of selected single items was undertaken with multivariable robust Poisson regression [29], for which the ordered responses of each item were collapsed into binary categories (Strongly agree/Agree versus all other responses), because in preliminary analysis the proportional odds assumption required for ordinal logistic regression [38] were not consistently met.

# Results

## Questionnaire completion

Among 1,159 persons who commenced the online survey, 1024 (88.4%) responded to at least one FDV-related item, and 914 (78.9%) completed all items. Non-completion was independently and inversely associated with highest level of attained education (relative risk of non-completion compared with university-educated respondents: Year 10 or below 2.01 [95% CI 1.30–3.10; Year 12 1.81 [95% CI 1.12–2.91]) but was not independently associated with age group, gender, Indigenous identification, country of birth, or income category.

## Sociodemographic characteristics

The sociodemographic characteristics of respondents who completed the survey are displayed in Table 1, based on the original (unweighted) data along with the raked weights used for all analyses. Seventy three percent of the respondents were female, 39.3% were university educated, 81.1% were Australian born and 64% were aged between 25–64 years. Although the

proportion of respondents who identified as Indigenous (n = 77; 8.4%) was similar to that reported in Census data for the Greater Geraldton population (9.7%), Indigenous compared with non-Indigenous (n = 837) respondents differed substantially in regard to the other modelled sociodemographic covariates.

## Knowledge, attitudes and personal experiences of FDV

In relation to overall responses to individual questionnaire items, there was nearly uniform recognition (94.9–96.7% of respondents) that each of the listed behaviours constituted family/domestic violence at least 'Sometimes'. All of the listed behaviours were recognised as 'Always' constituting FDV by a clear majority of respondents, although the proportions who identified certain emotionally abusive ('Repeatedly criticises partner to make them feel bad or useless': 78.5%) or coercive controlling behaviours ('Tries to control their partner by denying them money': 75.7%; 'Repeatedly keeps track of their partner or former partner's location, calls or activities': 72.8%) was substantially lower than the proportions who recognised physical acts or threats as 'Always' constituting FDV (>87%) (Fig 1A). In relation to knowledge of FDV causes, substance use (alcohol: 89.0%; illicit drugs: 91.5%) were most commonly identified, whereas factors related to traditional gender values were less consistently identified ('Men's loss of role and status': 62.6%; 'Men wanting to control women': 83.6%) (Fig 1B). Responses for items concerning attitudes to FDV indicated majority agreement for bystander acts ('I would tell a male friend when his behaviour is sexist': 80.5% Strongly agree/Agree; 'If I saw a male friend insulting or verbally abusing a woman he is in a relationship with I would do or say something to show I didn't approve': 85.2% Strongly agree/Agree). Contrastingly, 'Agree'/'Strongly Agree' responses were provided by only a minority (<25%) in relation to all nineteen of the FDV-supportive attitudes (Fig 1C). Substantial minorities of respondents reported having ever personally experienced various FDV behaviours, ranging from 15.4% in the case of threats 'to hurt other family members' to 43.8% of respondents who had ever experienced repeated criticism 'to make [them] feel bad or useless'. For all of the behaviours, reported experiences were far lower during the past six months than 'at any time in the past' (Fig 1D).

In exploratory factor analysis, differences in structure between unweighted and weighted datasets were negligible, and only the latter was used for regression and dominance analyses. The survey items on knowledge of FDV types loaded unambiguously onto a single factor, as did the items on personal experiences of FDV. However, the single factor extractable on knowledge of FDV causes included only three of eight items, and the exploratory factor analysis on FDV attitudes produced three factors that excluded five items (Table 2).

## Predictors of FDV-related perspectives

Independent sociodemographic predictors were determined for responses to the composite measures extracted with factor analysis (Table 3) and for individual items that could not be loaded onto a factor (Table 4). Female compared with male gender predicted higher knowledge of FDV types and of personal experiences as a victim of FDV, along with lower levels of FDV-supporting attitudes. Indigenous identification independently predicted lower knowledge of FDV types and higher levels of 'Domestic violence minimising' and 'Male chauvinism' attitudes but not personal experience of FDV. Respondents' age group was not significantly correlated with knowledge of FDV types, whereas respondents aged >65 years demonstrated higher levels of FDV-supporting attitudes than younger respondents, with the pattern across age-groups varying according to the age. Participants aged >65 years reported lower levels of personal experience of FDV than all younger age-groups. A university education predicted lower levels of FDV-supporting attitudes but was not significantly correlated with knowledge

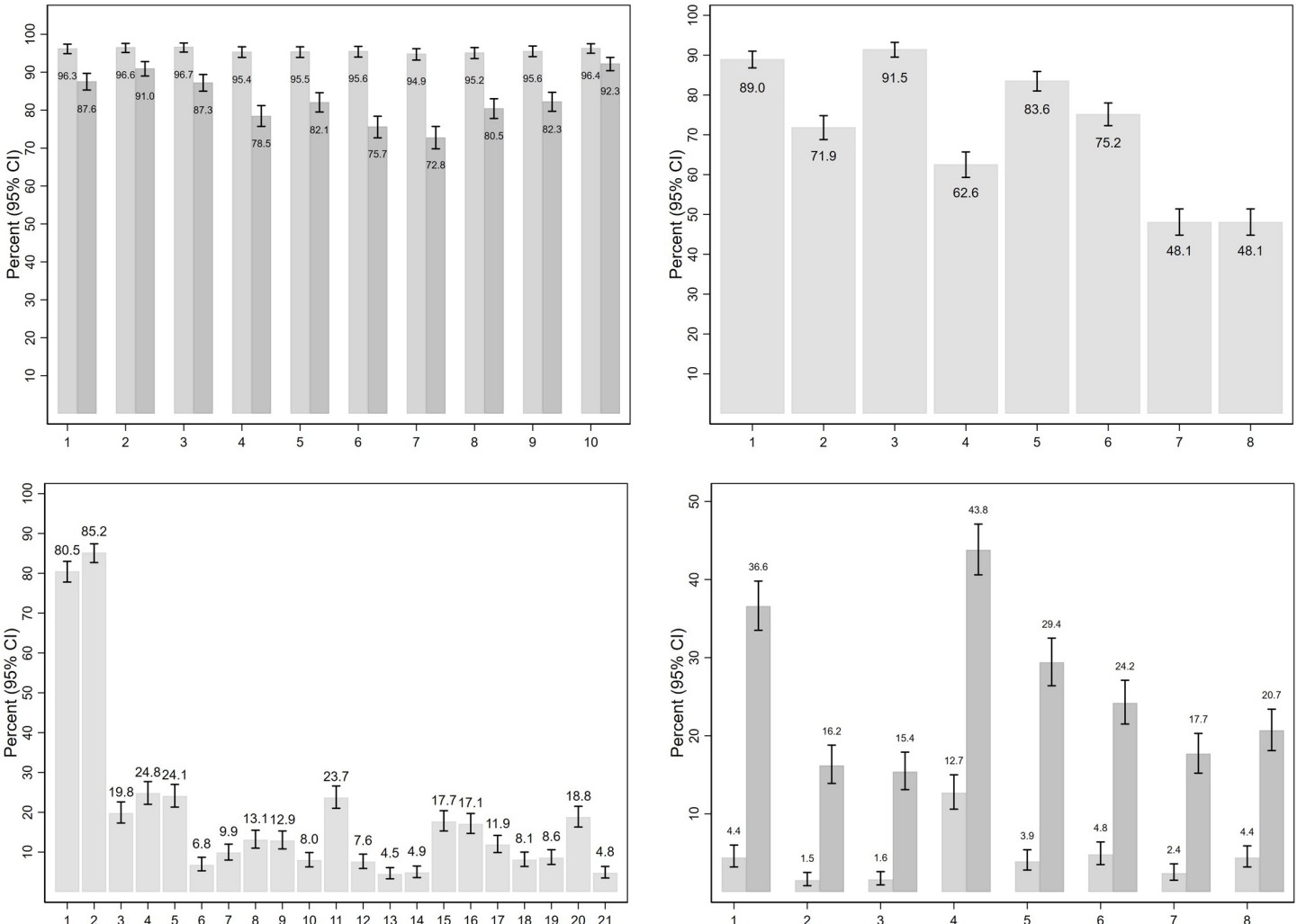

**Fig 1. A. Responses to individual items on knowledge of FDV behaviours (total sample, weighted data).** Do you consider the following behaviours to be family and domestic violence? When a person: 1 Slaps or pushes their partner to cause harm or fear; 2 Tries to scare or control their partner by threatening to hurt other family members; 3 Throws or smashes objects near their partner to frighten or threaten them; 4 Repeatedly criticises their partner to make them feel bad or useless; 5 Controls their partner's social life by preventing them from seeing family and friends; 6 Tries to control their partner by denying them money; 7 Repeatedly keeps track of their partner or former partner's location, calls or activities; 8 Repeatedly follows or watches a partner or former partner; 9 Repeatedly sends partner or former partner unwanted phone calls, emails, text messages and the like; 10 Forces their partner to have sex. Lighter grey shade: respondent considers that the behaviour is sometimes/ usually/always FVD. Darker grey shade: respondent considers that the behaviour is always FVD. Proportions and 95% confidence intervals were estimated with Fisher's exact test, then converted to percentage values. **B. Responses to individual items on knowledge of FDV causes–percent of respondents who Agree or Strongly Agree that the item is a cause (total sample, weighted data).** How much do you agree or disagree that the following cause family and domestic violence? 1 Alcohol; 2 Lack of employment; 3 Use of illegal drugs; 4 Men's loss of role and status; 5 Men wanting to control women; 6 Violence in the community; 7 Violence in the media (movies, video games etc.); 8 Pornography. Proportions and 95% confidence intervals were estimated with Fisher's exact test, then converted to percentage values. **C. Responses to individual items on attitudes towards FDV (total sample, weighted data)–percent who Agree or Strongly agree with each attitude.** How much do you agree or disagree with the following statements? 1 I would tell a male friend when his behaviour is sexist; 2 If I saw a male friend insulting or verbally abusing a woman he is in a relationship with I would do or say something to show I didn't approve; 3 If a person hits you, you should hit them back; 4 It is okay to hit children if they have done something wrong; 5 If people threaten my family / friends they deserve to get hurt; 6 Family and domestic violence is a private matter to be handled in the family; 7 Family and domestic violence can be excused if afterwards the person genuinely regrets what they have done; 8 If a woman reports abuse by her partner to outsiders it is shameful for her family; 9 Many women tend to exaggerate the problem of male violence; 10 If a woman is raped while she is drunk or affected by drugs she is at least partly responsible; 11 A woman who does not leave an abusive partner is partly responsible for the abuse continuing; 12 Women often say 'no' when they mean 'yes'; 13 If a woman wears revealing clothing, she is at least partly responsible for rape; 14 I'm more likely to listen to a man's opinion than a woman's; 15 It's ok for men to whistle at women while they are walking down the street; 16 I think there's no harm in men making sexist jokes about women when they are among their male friends; 17 Men should take control in relationships and be the head of the household; 18 Men make better political leaders than women; 19 In the workplace, men generally make more capable bosses than women; 20 Women seek to gain power by gaining control over men; 21 Violence by a man against his female partner can be excused if the offender is heavily intoxicated by alcohol. Proportions and confidence intervals estimated with Fisher's exact test, then converted to percentage values. **D. Responses to individual items on personal experiences of FDV (total sample, weighted data), percent.** When, if ever, was your most recent experience of any of these from someone you were in a relationship with (dating, de-facto, or married)? 1 Slapped, pushed or hit you; 2 Forced you to have sex; 3 Threatened to hurt other family members; 4 Repeatedly

criticised you to make you feel bad or useless; 5 Threw or smashed an object near you to cause fear; 6 Prevented you from seeing friends or family; 7 Tracked your location without your consent or monitored your phone calls and messages; 8 Repeatedly sent you unwanted phone calls, emails, text messages and the like. Lighter grey shade: respondent reports experience within the last six months. Darker grey shade: respondent reports experience at any time in the past. Proportions and confidence intervals were estimated with Fisher's exact test, then converted to percentage values.

of FDV types or of personal FDV experiences. Country of birth was not an independent predictor of FDV-related constructs except that the respondents born in non-Anglophone countries reported significantly lower support than other respondents for General Violence, and those born in Anglophone countries other than Australia reported lower levels than others of personal FDV experience. Income level was not an independent predictor of knowledge of FDV types or of attitudes supporting FDV, whereas respondents in higher income brackets reported lower levels of personal FDV experience.

There were few significant associations between sociodemographic subgroups of respondents and the individual items on knowledge of FDV causes that did not load onto any factor. Female respondents had higher agreement than males that men wanting to control women, violence in the media, and pornography are causes of FDV. Younger compared with older respondents and those receiving higher compared with lower incomes were less likely to agree that violence in the media is a cause of FDV. Respondents in the higher income brackets were less likely than those on lower incomes to identify pornography as a cause. No consistent pattern and few significant correlations were evident between sociodemographic subgroups and non-factoring individual items on attitudes towards FDV, except that respondents aged >65 years were more likely than all those aged <65 years to agree that 'it's ok for men to whistle at women while they are walking down the street'.

The analysis of relative importance in predicting responses to each construct of the sociodemographic attributes and other constructs is presented as Table 5. The 'Male Chauvinism' construct was of principal importance in determining responses to knowledge of FDV causes as well as to both of the other 'Attitude' constructs. Personal experience of FDV contributed extremely small proportions of the variance (0.000–0.003) of responses to any of the knowledge and attitude constructs.

## Discussion

This online survey conducted in outer regional Australia has provided local data on community understanding of and attitudes towards FDV, along with the novel incorporation into a single instrument of items on personal FDV experiences. Participation in the survey was substantial, with almost a thousand completions recorded. From this sizeable sample, it was possible to investigate higher-order constructs and to delineate independent sociodemographic predictors of responses to these constructs, except in relation to the items on knowledge of FDV causes and many of those on attitudes towards FDV, which factored poorly. A detailed descriptive account of individual LCAEVS item responses and their predictors, including item-by-item comparison with NCAS results where applicable, has been reported elsewhere (https://tinyurl.com/LCAEVSreport [39]). From the existing published literature on community perspectives towards FDV, the data most obviously suitable for comparison with LCAEVS are those from the Australian NCAS survey, from which many of the LCAEVS items were derived or adapted [25]. The constructs extracted from responses to the questionnaire were broadly in keeping with those from national findings albeit inevitably with some loss of loading sensitivity (i.e., the proportion of items loading satisfactorily onto at least one factor) [40] compared with the source instruments, perhaps an inevitable trade-off given the abridgement of included items.

**Table 3. Independent socio-demographic predictors of responses to composite measures.**

| | Knowledge—types | Knowledge—causes | Attitude 1 | Attitude 2 | Attitude 3 | Experience (victim) |
|---|---|---|---|---|---|---|
| | | | 'Domestic violence minimising' | 'Male chauvinism' | 'General violence' | |
| | Co-Efficient (95% CI) | Co-Efficient (95% CI) | Co-Efficient (95% CI) | Co-Efficient (95% CI) | Co-Efficient (95% CI) | Co-Efficient (95% CI) |
| **Gender** | | | | | | |
| **Male** | Reference | Reference | Reference | Reference | Reference | Reference |
| **Female** | 1.66 (0.53–2.79)† | -0.19 (-0.55–0.18) | -3.19 (-4.18 –-2.20)‡ | -2.51 (-3.1 –-1.93)‡ | -1.05 (-1.49 –-0.60)‡ | 1.70 (0.67–2.74)† |
| **Indigenous identification** | | | | | | |
| **non-Indigenous** | Reference | Reference | Reference | Reference | Reference | Reference |
| **Indigenous** | -3.62 (-6.49 –-0.76)* | -0.39 (-1.07–0.29) | 4.25 (1.61–6.89)† | 1.78 (0.53–3.02)† | 0.72 (-0.06–1.50) | -0.79 (-2.88–1.31) |
| **Age group** | | | | | | |
| **15–24 years** | -0.46 (-3.16–2.24) | -0.87 (-1.63 –-0.10)* | -1.88 (-4.47–0.72) | -0.83 (-2.26–0.60) | -1.20 (-2.33 –-0.08)* | 4.04 (1.43–6.65)† |
| **25–34 years** | 1.26 (-1.45–3.97) | -0.20 (-0.82–0.43) | -2.82 (-5.04 –-0.60)* | -0.41 (-1.58–0.76) | -1.71 (-2.71 –-0.71)† | 3.30 (1.55–5.04)‡ |
| **35–44 years** | 0.39 (-2.28–3.06) | -0.63 (-1.23 –-0.03)* | -3.25 (-5.31 –-1.19)† | -0.52 (-1.66–0.63) | -1.31 (-2.32 –-0.30)* | 4.08 (2.39–5.78)‡ |
| **45–54 years** | 0.78 (-1.94–3.50) | -0.58 (-1.22–0.06) | -3.34 (-5.42 –-1.25)† | -1.00 (-2.07–0.07) | -1.01 (-2.02–0.00) | 4.27 (2.48–6.07)‡ |
| **55–64 years** | 1.83 (-0.78–4.43) | -0.26 (-0.82–0.30) | -4.31 (-6.50 –-2.12)‡ | -1.29 (-2.39 –-0.18)* | -0.83 (-1.83–0.16) | 2.04 (0.76–3.32)† |
| **65+ years** | Reference | Reference | Reference | Reference | Reference | Reference |
| **Highest education** | | | | | | |
| **Year 10 or below** | -1.09 (-2.71–0.52) | 1.04 (0.58–1.49)‡ | 2.42 (0.84–3.99)† | 0.90 (0.05–1.75)* | 0.71 (0.07–1.35)* | 1.34 (-0.04–2.72) |
| **Year 12** | -0.36 (-1.61–0.90) | 0.52 (0.02–1.01)* | 1.93 (0.68–3.18)† | 0.69 (-0.11–1.49) | 1.01 (0.39–1.64)† | -0.87 (-2.09–0.35) |
| **Trade/ TAFE** | -0.93 (-1.91–0.05) | 0.45 (0.02–0.88)* | 1.21 (0.32–2.11)† | 1.09 (0.51–1.68)‡ | 0.95 (0.46–1.45)‡ | 0.41 (-0.43–1.26) |
| **University** | Reference | Reference | Reference | Reference | Reference | Reference |
| **Birth country** | | | | | | |
| **Australia** | Reference | Reference | Reference | Reference | Reference | Reference |
| **Non-Aus Anglo** | 0.23 (-1.59–2.05) | 0.27 (-0.35–0.90) | -0.70 (-1.96–0.56) | 0.14 (-0.55–0.82) | 0.27 (-0.47–1.02) | -1.90 (-2.81 –-1.00)‡ |
| **Other** | -0.57 (-3.36–2.21) | -0.11 (-0.96–0.75) | -0.21 (-3.41–2.99) | -1.18 (-2.56–0.21) | -1.34 (-2.44 –-0.23)* | -0.95 (-2.68–0.78) |
| **Income** | | | | | | |
| **<$25,000** | Reference | Reference | Reference | Reference | Reference | Reference |
| **$25,000–50,000** | 0.00 (-1.90–1.90) | 0.15 (-0.50–0.80) | -0.42 (-2.57–1.74) | 0.27 (-0.85–1.40) | 0.57 (-0.39–1.52) | -1.10 (-3.11–0.90) |
| **$51,000–100,000** | 0.04 (-2.21–2.30) | 0.36 (-0.22–0.94) | -1.81 (-3.79–0.16) | -0.48 (-1.44–0.48) | -0.18 (-1.02–0.66) | -2.20 (-3.90 –-0.49)* |
| **$101,000–150,000** | -0.39 (-2.27–1.48) | 0.27 (-0.39–0.93) | -0.79 (-2.99–1.41) | -0.12 (-1.20–0.96) | -0.12 (-1.02–0.78) | -3.20 (-5.17 –-1.24)† |
| **Over $150,000** | -0.27 (-2.35–1.81) | -0.38 (-1.12–0.36) | -1.46 (-3.38–0.45) | -0.25 (-1.32–0.83) | 0.28 (-0.65–1.20) | -3.45 (-5.58 –-1.31)† |

Multiple linear regression model with robust standard errors, comprising all covariates listed in table, rake-weighted dataset

\* $p < 0.05$

† $p < 0.01$

‡ $p < 0.001$

Coefficients indicate the mean increment of responses from stated sub-group, compared with reference sub-group, across 33-point composite scale (minimum possible score [least recent/ fewest types of experiences] = 8; maximum possible score [most recent/ most types of experiences] = 40)

CI: confidence interval

Age group: 65+ years age group used as reference category because of outlier status noted across preliminary analyses

Highest education level attained: University-educated group used as reference category because of outlier status noted across preliminary analyses

Sociodemographic patterns of response to the online survey were notably characterised by the markedly disproportionate participation of women as well as substantial under-representation of persons from extremes of age and over-representation of those with a university

**Table 4. Independent socio-demographic predictors of responses to individual items without adequate factor loading [*].**

| | | Q19-4 | Q19-5 | Q19-6 | Q19-7 | Q19-8 |
|---|---|---|---|---|---|---|
| | | Risk ratio (95% CI) | Risk ratio (95% CI) | Risk ratio (95% CI) | Risk ratio (95% CI) | Risk ratio (95% CI) |
| **Gender** | Male | Reference | Reference | Reference | Reference | Reference |
| | Female | 1.09 (0.94–1.26) | 1.10 (1.01–1.19)[*] | 0.95 (0.86–1.05) | 1.24 (1.03–1.49)[*] | 1.40 (1.15–1.70)[†] |
| **Indigenous identity** | non-Indigenous | Reference | Reference | Reference | Reference | Reference |
| | Indigenous | 1.03 (0.80–1.31) | 0.86 (0.72–1.02) | 1.08 (0.93–1.27) | 1.37 (1.05–1.78)[*] | 1.10 (0.81–1.48) |
| **Age group** | 15–24 years | 0.66 (0.47–0.92)[*] | 1.03 (0.87–1.22) | 0.79 (0.63–0.98)[*] | 0.46 (0.29–0.71)[†] | 0.82 (0.55–1.21) |
| | 25–34 years | 0.83 (0.64–1.08) | 1.04 (0.88–1.24) | 0.90 (0.76–1.07) | 0.52 (0.37–0.74)[‡] | 0.68 (0.47–0.98)[*] |
| | 35–44 years | 0.84 (0.66–1.07) | 0.99 (0.83–1.17) | 0.91 (0.77–1.07) | 0.55 (0.40–0.77)[‡] | 0.76 (0.54–1.08) |
| | 45–54 years | 0.77 (0.61–0.98)[*] | 0.98 (0.82–1.17) | 0.92 (0.78–1.07) | 0.78 (0.59–1.03) | 0.79 (0.57–1.08) |
| | 55–64 years | 0.83 (0.65–1.05) | 0.98 (0.82–1.17) | 0.91 (0.77–1.06) | 0.85 (0.65–1.11) | 0.95 (0.70–1.30) |
| | 65+ years | Reference | Reference | Reference | Reference | Reference |
| **Highest education** | Year 10 or below | 1.20 (1.00–1.44) | 0.99 (0.88–1.12) | 0.97 (0.85–1.11) | 0.87 (0.68–1.11) | 0.84 (0.66–1.07) |
| | Year 12 | 1.08 (0.88–1.32) | 1.02 (0.92–1.13) | 0.95 (0.82–1.09) | 0.80 (0.61–1.06) | 0.69 (0.52–0.92)[*] |
| | Trade/ TAFE | 1.10 (0.96–1.27) | 1.01 (0.93–1.10) | 1.00 (0.91–1.11) | 0.92 (0.77–1.10) | 0.91 (0.77–1.09) |
| | University | Reference | Reference | Reference | Reference | Reference |
| **Birth country** | Australia | Reference | Reference | Reference | Reference | Reference |
| | Non-Aus Anglo | 1.07 (0.89–1.28) | 1.08 (0.98–1.20) | 0.98 (0.85–1.13) | 0.84 (0.62–1.12) | 0.71 (0.50–1.00)[*] |
| | Other | 1.06 (0.81–1.38) | 1.14 (1.04–1.24)[†] | 0.83 (0.61–1.12) | 1.17 (0.84–1.63) | 1.48 (1.14–1.91)[†] |
| **Income** | <$25,000 | Reference | Reference | Reference | Reference | Reference |
| | $25,000–50,000 | 0.85 (0.67–1.08) | 0.93 (0.81–1.07) | 1.13 (0.98–1.30) | 0.85 (0.64–1.12) | 0.78 (0.57–1.05) |
| | $51,000–100,000 | 0.88 (0.72–1.07) | 0.96 (0.86–1.08) | 0.97 (0.83–1.14) | 0.76 (0.60–0.98)[*] | 0.82 (0.63–1.05) |
| | $101,000–150,000 | 1.01 (0.81–1.26) | 0.89 (0.77–1.02) | 0.96 (0.78–1.17) | 0.73 (0.52–1.02) | 0.71 (0.51–0.99)[*] |
| | Over $150,000 | 0.92 (0.72–1.17) | 0.92 (0.80–1.07) | 0.97 (0.79–1.20) | 0.60 (0.43–0.85)[†] | 0.69 (0.50–0.96)[*] |
| | | Q20-1 | Q20-2 | Q20-3 | Q20-4 | Q20-5 |
| | | Risk ratio (95% CI) | Risk ratio (95% CI) | Risk ratio (95% CI) | Risk ratio (95% CI) | Risk ratio (95% CI) |
| **Gender** | Male | Reference | Reference | Reference | Reference | Reference |
| | Female | 1.15 (1.05–1.26)[†] | 1.00 (0.93–1.08) | 0.19 (0.07–0.53)[†] | 0.80 (0.54–1.19) | 0.28 (0.19–0.42)[‡] |
| **Indigenous identity** | non-Indigenous | Reference | Reference | Reference | Reference | Reference |
| | Indigenous | 1.13 (0.99–1.30) | 0.94 (0.81–1.10) | 3.28 (1.02–10.56)[*] | 1.57 (0.85–2.90) | 1.32 (0.72–2.41) |
| **Age group** | 15–24 years | 1.23 (0.97–1.56) | 1.26 (1.03–1.53)[*] | 1.65 (0.13–20.83) | 0.25 (0.12–0.54)[‡] | 0.96 (0.41–2.27) |
| | 25–34 years | 1.16 (0.93–1.45) | 1.11 (0.92–1.34) | 1.15 (0.14–9.38) | 0.28 (0.13–0.59)[†] | 1.29 (0.60–2.77) |
| | 35–44 years | 1.05 (0.83–1.31) | 1.17 (0.97–1.40) | 1.74 (0.24–12.81) | 0.34 (0.18–0.65)[†] | 0.89 (0.41–1.92) |
| | 45–54 years | 1.02 (0.82–1.27) | 1.14 (0.96–1.36) | 1.82 (0.23–14.33) | 0.35 (0.20–0.62)[‡] | 1.04 (0.50–2.17) |
| | 55–64 years | 1.08 (0.87–1.35) | 1.10 (0.90–1.33) | 1.44 (0.20–10.44) | 0.30 (0.16–0.56)[‡] | 0.66 (0.30–1.46) |
| | 65+ years | Reference | Reference | Reference | Reference | Reference |
| **Highest education** | Year 10 or below | 0.97 (0.86–1.09) | 0.90 (0.80–1.01) | 3.59 (1.03–12.50)[*] | 2.06 (1.20–3.53)[†] | 1.71 (0.94–3.08) |
| | Year 12 | 0.89 (0.78–1.03) | 0.98 (0.88–1.09) | 2.22 (0.46–10.72) | 1.11 (0.49–2.47) | 1.23 (0.61–2.48) |
| | Trade/ TAFE | 0.97 (0.89–1.05) | 1.07 (1.00–1.14) | 2.94 (0.83–10.42) | 1.72 (1.02–2.92)[*] | 1.59 (0.94–2.69) |
| | University | Reference | Reference | Reference | Reference | Reference |
| **Birth country** | Australia | Reference | Reference | Reference | Reference | Reference |
| | Non-Aus Anglo | 1.20 (1.06–1.36)[†] | 1.11 (1.00–1.22)[*] | 1.32 (0.28–6.21) | 0.58 (0.27–1.24) | 0.77 (0.33–1.78) |
| | Other | 1.00 (0.77–1.29) | 1.04 (0.87–1.23) | 1.50 (0.13–16.85) | 0.50 (0.09–2.82) | 0.77 (0.24–2.45) |

(*Continued*)

**Table 4.** (Continued)

| Income | <$25,000 | Reference | Reference | Reference | Reference | Reference |
|---|---|---|---|---|---|---|
| | **$25,000–50,000** | 1.25 (1.02–1.52)[*] | 1.20 (1.01–1.43)[*] | 0.29 (0.03–2.44) | 0.90 (0.55–1.48) | 1.52 (0.78–2.99) |
| | **$51,000–100,000** | 1.09 (0.91–1.31) | 1.08 (0.92–1.26) | 0.73 (0.21–2.56) | 0.48 (0.27–0.86)[*] | 0.79 (0.40–1.59) |
| | **$101,000–150,000** | 1.12 (0.93–1.35) | 1.04 (0.88–1.22) | 0.37 (0.09–1.53) | 0.54 (0.26–1.10) | 0.92 (0.43–1.95) |
| | **Over $150,000** | 1.19 (0.98–1.44) | 1.06 (0.89–1.26) | 0.87 (0.18–4.09) | 0.79 (0.38–1.65) | 1.09 (0.47–2.55) |

[*]Multivariable robust Poisson regression models, weighted

Questions: How much do you agree or disagree that the following cause family and domestic violence?

19–4 Men's loss of role and status

19–5 Men wanting to control women

19–6 Violence in the community

19–7 Violence in the media (movies, video games etc.)

19–8 Pornography

How much do you agree or disagree with the following statements?

20–1 I would tell a male friend when his behaviour is sexist

20–2 If I saw a male friend insulting or verbally abusing a woman he is in a relationship with I would do or say something to show I didn't approve

20–3 I'm more likely to listen to a man's opinion than a woman's

20–4 It's ok for men to whistle at women while they are walking down the street.

20–5 I think there's no harm in men making sexist jokes about women when they are among their male friends

education. The LCAEVS respondent profile was similar to that of NCAS [25], and is typical of community health surveys in general [41, 42]. The sensitive nature of FDV likely accentuates non-response [43]. Non-completion of LCAEVS by a substantial proportion of respondents (~21%) who started the survey is in keeping those recorded for other web-based surveys [44], but may have accentuated selection bias. It is not possible to determine the extent to which participation in or completion of LCAEVS was influenced by the mode of delivery, length, sensitivity of the overall topic, or item content and lexical complexity. The web-based delivery mode constrains comparability of participation and completion with both NCAS (random-digit dialling telephone interview with a high proportion of refusals) [26] and PSS (face-to-face interview) [27], which each have differing selection biases that are remediated only partially by weighting.

The results indicate relatively less community awareness that emotional abuse and coercive control—compared with physical acts or threats and of forced sex—are forms of violence. LCAEVS and NCAS respondents both had high overall recognition of the FDV continuum but more consistently recognised incidents of obvious physical violence and forced sex than controlling behaviours and other forms of emotional abuse as forms of FDV [25] (p6). In part, this likely reflects justice system responses [45] and media reporting [46]. Knowledge of causal factors for FDV in LCAEVS (highest for substance abuse, lowest for media violence and pornography, with traditional gender roles intermediate) did not strictly replicate the delineation of causal attribution to 'individual' and 'broader social' factors in NCAS [25] (pp 58–60). The findings common to these Australian surveys are broadly concordant with those conducted in similar countries. Internationally, sexist values and acceptance of violence in general are positively associated with acceptance of intimate partner violence [47]. Based on a random digit dialling telephone interview survey in New York State, in the context of intimate relationships, acts of physical violence and forced sex were more consistently considered domestic violence by respondents (n = 1200) than were insulting comments [48]. In the latter survey, similar to

**Table 5. Relative importance\* of covariates in predicting responses to composite measures.**

| | $R^2$ | Rel. $R^2$ | Rank | | $R^2$ | Rel. $R^2$ | Rank | | $R^2$ | Rel. $R^2$ | Rank |
|---|---|---|---|---|---|---|---|---|---|---|---|
| **Knowledge composite measure** | | | | **Attitudes composite measure 1** | | | | **Attitudes composite measure 2** | | | |
| Attitude CM 2 | 0.025 | 0.251 | 1 | Attitude CM 2 | 0.261 | 0.513 | 1 | Attitude CM 1 | 0.262 | 0.505 | 1 |
| Indigenous identity | 0.022 | 0.219 | 2 | Attitude CM 3 | 0.055 | 0.107 | 2 | Gender | 0.089 | 0.171 | 2 |
| Attitude CM 3 | 0.013 | 0.133 | 3 | Age group | 0.049 | 0.097 | 3 | Attitude CM 3 | 0.085 | 0.164 | 3 |
| Age group | 0.011 | 0.113 | 4 | Gender | 0.044 | 0.086 | 4 | Knowledge CM | 0.021 | 0.040 | 4 |
| Attitude CM 1 | 0.011 | 0.108 | 5 | Income | 0.032 | 0.063 | 5 | Age group | 0.020 | 0.038 | 5 |
| Gender | 0.010 | 0.105 | 6 | Indigenous identity | 0.029 | 0.057 | 6 | Indigenous identity | 0.014 | 0.027 | 6 |
| Highest education | 0.004 | 0.035 | 7 | Highest education | 0.024 | 0.047 | 7 | Income | 0.010 | 0.020 | 7 |
| Income | 0.002 | 0.022 | 8 | Knowledge CM | 0.010 | 0.019 | 8 | Highest education | 0.010 | 0.019 | 8 |
| Birth country | 0.001 | 0.012 | 9 | Birth country | 0.004 | 0.008 | 9 | Birth country | 0.005 | 0.010 | 9 |
| Experience CM | 0.000 | 0.002 | 10 | Experience CM | 0.002 | 0.004 | 10 | Experience CM | 0.003 | 0.006 | 10 |
| OVERALL | 0.099 | 1.000 | - | OVERALL | 0.508 | 1.000 | - | OVERALL | 0.518 | 1.000 | - |
| **Attitudes composite measure 3 'General violence'** | | | | **'Experience' composite measure** | | | | | | | |
| Attitude CM 2 | 0.095 | 0.347 | 1 | Age group | 0.058 | 0.389 | 1 | | | | |
| Attitude CM 1 | 0.059 | 0.215 | 2 | Gender | 0.027 | 0.184 | 2 | | | | |
| Age group | 0.039 | 0.143 | 3 | Income | 0.022 | 0.149 | 3 | | | | |
| Gender | 0.023 | 0.086 | 4 | Birth country | 0.015 | 0.103 | 4 | | | | |
| Income | 0.018 | 0.065 | 5 | Highest education | 0.014 | 0.096 | 5 | | | | |
| Knowledge CM | 0.012 | 0.045 | 6 | Attitude CM 2 | 0.005 | 0.034 | 6 | | | | |
| Highest education | 0.012 | 0.044 | 7 | Attitude CM 1 | 0.003 | 0.020 | 7 | | | | |
| Birth country | 0.011 | 0.041 | 8 | Attitude CM 3 | 0.003 | 0.019 | 8 | | | | |
| Experience CM | 0.003 | 0.009 | 9 | Indigenous identity | 0.001 | 0.006 | 9 | | | | |
| Indigenous identity | 0.002 | 0.007 | 10 | Knowledge CM | 0.000 | 0.002 | 10 | | | | |
| OVERALL | 0.273 | 1.000 | - | OVERALL | 0.149 | 1.000 | - | | | | |

\*Based on dominance analysis model comprising all listed covariates (sociodemographic characteristics of respondents and other constructs extracted in factor analysis)

Rel. $R^2$: Relative $R^2$

Knowledge construct comprises all ten knowledge items in survey

Attitude construct 2: 'Male chauvinism' (composite of four survey items)

Attitude construct 3: 'General violence' (composite of three survey items)

Experience construct comprises all eight experience items in survey

the NCAS findings, respondents attributed domestic violence more often to individual-level than broader social and cultural causes [49].

LCAEVS respondents' attitudes towards FDV indicated overwhelming majority support for 'bystander' intentions (i.e., confronting a male friend for sexist comments or verbally abuse/insulting behaviour towards a woman partner) and only minority agreement (4.5–24.8%) with various attitudes promoting FDV. The latter findings are encouraging and have plausible claim to validity, considering that social-desirability bias tends to be diminished in responses to questionnaires delivered online in comparison with those obtained through telephone or face-to-face interview [50].

Exploratory factor analysis of LCAEVS responses produced a similar albeit incomplete version of the constructs produced from the more extensive list of comparable items in the NCAS source instrument. An unambiguous factor structure could not be extracted from the sample data for most of the eight items addressing knowledge of possible causes of FDV, with only a single composite measure comprising three items (alcohol, illegal drugs and unemployment)

accepted as causal in FDV significantly more often by respondents without than those with a university education.

In relation to respondents' attitudes towards gender equality, LCAEVS results did not provide the conceptual precision that has been feasible using NCAS data to identify and investigate multiple distinct themes: promotion of rigid or stereotyped gender roles, support for undermining women's independence differentially in public and private life settings, condoning of male aggression and disrespect towards women and denial of gender equality. Similarly, in relation to a conceptual framework for violence-supportive attitudes, NCAS enabled drilling down farther than LCAEVS in the recognition of four distinct themes: excusing perpetrators, denying or downplaying the gravity of FDV, mistrust of women's reporting of violence, and disregard towards the need for consent in sexual relations. From LCAEVs, the eight-item construct reflecting an attitude that minimises FDV received significantly greater agreement among male compared with female respondents, from Indigenous compared with non-Indigenous respondents, from respondents aged >65 years compared with those from most younger age groups, and from respondents without compared to those with a university education. Similar patterns of sociodemographic predictors were evident for the two other composite measures dealing with respondents' attitudes towards FDV, although the 'male chauvinism' construct was less clearly predicted by respondents' age, and the 'general violence' construct was not significantly associated with Indigenous identification. To some extent, LCAEVS and NCAS results were similar in eliciting associations between violence- and gender inequality-promoting attitudes and sociodemographic attributes of respondents such as gender, age, education and income. However, given the far larger sample size than LCAEVS, NCAS data (N = 17,542) provided greater precision in subgroup analysis. LCAEVS findings of higher levels of violence-supporting attitudes among Indigenous than non-Indigenous respondents are broadly compatible with NCAS data, in which there were no significant nationwide differences between Indigenous and non-Indigenous respondents on attitude measures, but higher endorsement of such attitudes was evident among Indigenous respondents from inner and outer regional areas compared with those living in major cities [51] (pp 35–37). In both surveys, caution is warranted in interpretation of findings for Indigenous populations, given the sample size and sampling artefacts (i.e., the unrepresentative participation of Indigenous participants in relation to measured sociodemographic covariates).

The substantial proportion of respondents identifying personal experience at any time in their life as a victim of FDV in its various forms (most commonly reported 'repeated criticism' or being 'slapped, pushed or hit') is concerning, although these data must be interpreted with caution, given that respondents' decisions to participate in the survey may have been influenced by such experiences. Sociodemographic determinants of the composite measure capturing breadth and recency of experience as a victim of FDV are broadly concordant with national data from Australia [7] and other similar countries [52]: experiences were reported at significantly higher levels by female than male respondents, by younger respondents compared with over-65s, and by those reporting lower incomes, though, interestingly given their known higher rates of FDV [7], not by Indigenous compared with non-Indigenous respondents. As with NCAS, there are inevitably caveats on comparisons between LCAEVS results regarding personal experiences of FDV with those of the nationwide PSS [7], as the two instruments differed considerably in sampling strategy and mode of delivery, as well as in item content. At face value, the 43.8% of LCAEVS respondents who reported a lifetime experience of having been repeatedly criticised by a partner 'to make [them] feel bad or useless' appears to exceed the 'one in four women and one in six men' in the PSS sample who reported having experienced 'partner emotional abuse since the age of 15'. In both surveys, physical violence was

reported much more commonly than forced sexual activity, although in PSS data—unlike LCAEVS—actual and threatened acts of both types of violence are conflated in reporting.

Also with direct bearing on interpretation and contextualisation of the LCAEVS results is the similarly conceived *Domestic Violence in Wagga Wagga—Community Attitudes Survey* [28], which was conducted in the New South Wales inner regional city of Wagga Wagga (population ~ 62,000) [53] in 2016 and repeated in 2021 [54]. Like LCAEVS, the Wagga Wagga instrument was adapted from NCAS, and was administered through a combination of hardcopy and online participation. As with LCAEVS, poorer knowledge of the FDV behaviour continuum and attitudes supporting traditional gender roles and FDV were disproportionately evident among males, older respondents and those with lower education levels, as well as Indigenous respondents in the Wagga Wagga survey.

Finally, the relative importance among predictors of the various constructs must be interpreted with caution, given the susceptibility of dominance analysis to sampling biases and measurement errors and to model specification [55]. However, several clear patterns emerged from this analysis. The 'male chauvinism' construct was of high relative importance in accounting for responses both to knowledge of the FDV continuum and to the other attitudinal constructs, underscoring the centrality of traditional gender norms and their interaction with other core determinants of FDV [56].

Notably, personal experience of FDV was of negligible importance in predicting responses on knowledge of and attitudes towards FDV, and conversely, sociodemographic characteristics (especially age group, and also gender, income, birth country and education) were of substantially greater importance relative to the knowledge and attitude response constructs in predicting responses on personal experiences. As far as we can establish, this finding of non-relatedness between personal experience and knowledge and attitudes in the community setting is novel, while we acknowledge that it is both counterintuitive and is informed only obliquely by the literature from relevant workforce and student settings, in which there are conflicting findings on the influence of personal experience of FDV on knowledge and attitudes [57–59].

## Strengths and limitations

The large sample reflects considerable efforts in promoting community participation and provided for robust analysis but was unrepresentative of the local population, a problem remediated only partially by weighting. The sociodemographic imbalance in responses to LCAEVS is typical of surveys dealing with FDV [26, 60, 61]. The development of LCAEVS, based on combining and adapting items from pre-existing instruments inevitably engendered trade-offs in its aim to optimise participation and completion of the survey by reducing respondent burden against construct extraction. Despite these shortcomings, local data produced in the survey have been very useful for engaging local groups in discussions about FDV and its prevention.

## Conclusions

While the fundamental drivers of FDV are universal, how FDV is manifest and perceived is likely to be shaped uniquely in each community by local demographic and sociocultural determinants, even in a relatively homogeneous country such as Australia. The results of LCAEVS, along with those of the *Domestic Violence in Wagga Wagga—Community Attitudes Survey*, indicate that the demographic groups likely to hold attitudes supporting FDV in outer and inner regional Australia resemble those encountered nationwide. This survey provided local information to inform local community FDV primary prevention strategies, albeit with some loss of detail in demonstrated attitudinal patterns compared with those evident from

established national instruments. To some extent, the LCAEVS findings reinforce in a local context the preventive action recommended in NCAS targeting attitudes prevalent in particular community subgroups, including men and boys, older people and those experiencing various forms of disadvantage. For the purposes of primary prevention, in this regional setting as elsewhere, there is need to promote gender equality and to enhance public understanding of the role of traditional gender values such as male chauvinism and entitlement as primary drivers of FDV. Given the investment in infrastructure required to deliver NCAS at a national level, it should be considered whether there is a need for ANROWS to offer oversampling in some smaller regional centres that might circumvent the need for local surveys and allow for robust comparisons with national data that could be repeated over time. The novel and unexpected finding of negligible correlation between recency and breadth of personal experience as a victim of FDV and knowledge of and attitudes towards FDV is of potential importance for future interventions among persons directly affected.

## Acknowledgments

We thank Associate Professor Kevin Murray for statistical advice on weighting methods and Professor Colleen Fisher for advice and support related to this research.

## Author Contributions

**Conceptualization:** Heath S. Greville, Sandra C. Thompson.

**Data curation:** John A. Woods.

**Formal analysis:** John A. Woods.

**Funding acquisition:** Monica C. Moran, Barbara Nattabi, Karen E. Martin, Sandra C. Thompson.

**Investigation:** Heath S. Greville, Monica C. Moran, Barbara Nattabi, Karen E. Martin, Sandra C. Thompson.

**Methodology:** John A. Woods, Andrew C. Ward, Sandra C. Thompson.

**Project administration:** Heath S. Greville, Monica C. Moran, Sandra C. Thompson.

**Supervision:** Sandra C. Thompson.

**Writing – original draft:** John A. Woods.

**Writing – review & editing:** Andrew C. Ward, Heath S. Greville, Monica C. Moran, Barbara Nattabi, Karen E. Martin, Sandra C. Thompson.

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
