## [Decision Letter · Decision Letter 0]

3 Mar 2023

PONE-D-22-35475Measuring for primary prevention: an online survey of local community perspectives on family and domestic violence in regional AustraliaPLOS ONE

Dear Dr. Woods,

Thank you for submitting your manuscript to PLOS ONE. After careful consideration, we feel that it has merit but does not fully meet PLOS ONE’s publication criteria as it currently stands. Therefore, we invite you to submit a revised version of the manuscript that addresses the points raised during the review process.

We look forward to receiving your revised manuscript.

Kind regards,

Jianhong Zhou

Staff Editor

PLOS ONE

Journal Requirements:

2. We note you have included a table to which you do not refer in the text of your manuscript. Please ensure that you refer to Table 5 in your text; if accepted, production will need this reference to link the reader to the Table.

3. Please include a copy of Table 6 which you refer to in your text on page 26.

Reviewers' comments:

Reviewer's Responses to Questions

**Comments to the Author**

1. Is the manuscript technically sound, and do the data support the conclusions?

Reviewer #1: Yes

Reviewer #2: Yes

2. Has the statistical analysis been performed appropriately and rigorously? 

Reviewer #1: Yes

Reviewer #2: Yes

3. Have the authors made all data underlying the findings in their manuscript fully available?

Reviewer #1: Yes

Reviewer #2: No

4. Is the manuscript presented in an intelligible fashion and written in standard English?

Reviewer #1: Yes

Reviewer #2: Yes

5. Review Comments to the Author

Reviewer #1: The Research work is well carried out with proper survey

The results presented are satisfactory. there is need to promote gender equality and to enhance public understanding

of the role of traditional gender values such as male chauvinism and entitlement as primary drivers of FDV is well presented

The same can be carried out for other socially important metrics

Reviewer #2: Paper Title: " Measuring for primary prevention: an online survey of local community perspectives on family and domestic violence in regional Australia"

Paper ID: PONE-D-22-35475

Comments:

This paper proposes an effective way of analyzing the family and domestic violence (FDV) in regional Australia. It is socially relevant to the current scenario. The authors developed questionnaire-based survey to investigate FDV. The questionnaire comprised items on the sociodemographic characteristics of respondents in addition to FDV-related items. The paper is well written. Statistical analysis was performed in a well manner based on the reports from Local Community Attitudes and Exposure to Violence Survey (LCAEVS) and compared with the results of National Community Attitudes to Violence Against Women Survey (NCAS). Exploratory Factor Analysis was performed based on the domain (Knowledge of FDV types and causes, attitudes, personal experiences). This paper needs a minor revision before publication.

1. The analysis of survey data was performed in an effective manner and tabulated. (Table 1- Characteristics of survey respondents, Table 2- Exploratory factor analysis, Table 3- Multiple linear regression, Table 4- Multivariable robust Poisson regression, Table 5 - Dominance Analysis).

2. Data were weighted in respect to the respondents' self-reported gender, highest education level, Indigenous identifier and age group using the ipfweight command in Stata. Any temporal factor has been considered for weight assignment?.

3. Graphs such as Pie charts, Scatter plots and histograms are the most visually appealing survey data analysis methods. Only one type of plot (Fig 1a to Fig 1c) is utilized for visualization. If possible additional plots can be added.

Other minor issues are,

1. Check quotes

Uniform quotes pattern should be used. Straight quotes and curly quotes are used in mixed manner.

2. Check references

Uniform formatting should be followed for references. DOIs are missing for more references. Some references are having URLs. References can also be listed in alphabetical order.

6. PLOS authors have the option to publish the peer review history of their article (what does this mean?). If published, this will include your full peer review and any attached files.

Reviewer #1: No

Reviewer #2: No

---

## [Author Response · Author response to Decision Letter 0]

21 Mar 2023

Response to Reviewers

Dear Editor and Reviewers,

Thank you for your appraisal of our manuscript.

In the revised manuscript, we have made the following copy-editing corrections to minor errors that we ourselves have detected, as well as the error identified by the Editor, in which the citation of ‘Table 5’ in the main text had been misnumbered as ‘Table 6’. (There is actually no Table 6, and we apologise to the Editor for the error in the original submission.)

Line 83: Removal of space between reference [22] and full-stop

Line 132: Corrected spelling of ‘web chat’ 

Line 191: Corrected spelling of ‘proportional’ 

Line 227: Corrected Figure citation in text (‘Fig 1B’’ rather than Fig 1C’)

Line 232: Corrected spelling of ‘Substantial’

Line 250: Insertion of missing word ‘were’

Line 257: Insertion of missing word ‘were’

Line 288: Insertion of missing word ‘were’

Line 297 (within Table 2): Addition of full-stop to abbreviation ‘etc.’

Line 317: Addition of comma

Line 354 (within Table 4 legend): Addition of full-stop to abbreviation ‘etc.’ 

Line 363: Correction of table number (to ‘Table 5’), as directed by the Editor

Lines 365/366: Pluralisation of ‘proportions’

Line 585-586: Removal of superfluous PMID from reference #18

Line 616: Addition of missing DOI from journal reference #29

Line 630: presentation of full DOI not as URL, in accordance with journal style

Line 632: presentation of full DOI not as URL, in accordance with journal style 

Line 634: presentation of full DOI not as URL, in accordance with journal style

Reviewer 1 has recommended no revisions.

Our point-by-point responses to Reviewer 2 are provided below.

This paper needs a minor revision before publication. 

1. The analysis of survey data was performed in an effective manner and tabulated. (Table 1- Characteristics of survey respondents, Table 2- Exploratory factor analysis, Table 3- Multiple linear regression, Table 4- Multivariable robust Poisson regression, Table 5 - Dominance Analysis). 

Thank you.

Note that the incorrect numbering of tables within the text (‘Table 6’ instead of ‘Table 5’), identified by the Editor, has been corrected in the revised manuscript (line 366).

2. Data were weighted in respect to the respondents' self-reported gender, highest education level, Indigenous identifier and age group using the ipfweight command in Stata. Any temporal factor has been considered for weight assignment?

We recognise that the accuracy of sample weighting is dependent on the temporal proximity of data acquisition from the survey sample and the source population, and that time lags in data collection potentially give rise to weighting inaccuracies. 

The LCAEVS survey data presented in this manuscript are cross-sectional, with responses collected over a single six-week time window: 10 Oct 2019–30 Nov 2019. 

For our study, weighting was feasible only in relation to demographic indices of the local source population that were available in the public domain from national Census surveys. The population data for all assigned weights were derived from the (then) most recent national Census, which had been conducted in 2016, i.e., three years prior to LCAEVS. In response to the reviewer’s comments, we have examined data for the local (Greater Geraldton) area from the more recent 2021 Census, which are newly published and were not available at the time of analysis. The population proportions of all demographic indices used for our weighting have remained very stable between the two Census surveys (e.g., 2016: Male/Female = 49.6%:50.4%, Indigenous 9.7% versus 2021: Male/Female = 49.4%/50.6%, Indigenous 9.7%), so there is little potential for temporal bias in the weighting.

We presume that Reviewer 2’s response to Reviewer Question 3 (Have the authors made all data underlying the findings in their manuscript fully available? ‘No’) pertains to this point, as it is not otherwise mentioned in the Reviewer’s comments. 

3. Graphs such as Pie charts, Scatter plots and histograms are the most visually appealing survey data analysis methods. Only one type of plot (Fig 1a to Fig 1c) is utilized for visualization. If possible additional plots can be added. 

We agree in principle that a variety of graphical styles might enhance the presentation of results, and have considered various options for providing additional graphs.

Our core analytic aims were to investigate (i) the factor structure of responses to items in our survey and (ii) the independent associations (using multiple regression models) between these constructs and various demographic attributes of the surveyed population. The results of these analyses are suited to presentation in tables rather than figures. Consequently, our presentation of results is centred on tables. For the sake of completeness, we added bar charts (Figure 1 A–D) displaying percentage responses to individual survey items. The independent/predictor variables in our multiple regression models are categorical rather than numeric, and thus cannot be displayed as numeric-axis graphs such as scatter plots and histograms.

We considered that the dominance analysis (‘proportional contribution’) data can be presented graphically as an alternative to tables. Indeed, the national NCAS upon which we have modelled our research incorporated bar charts to this end (http://tinyurl.com/NCAS2017; e.g., Figure 12-1 and 12-2, pp 105–106). However, the addition of more bar charts would not enhance the diversity of graphical presentation of results, which is the purpose of the reviewer’s recommendation. Although there are published precedents for presenting dominance analysis data graphically as pie charts rather than bar charts, pie charts would be less readable than bar charts for our particular dominance analysis data, considering that some covariates account for minuscule proportions of explained variance. We consider that these data are most readably presented in table form.

In summary, therefore, our preference is to maintain the modes of data presentation in our original submission.

Other minor issues are:

1. Check quotes 

Uniform quotes pattern should be used. Straight quotes and curly quotes are used in mixed manner.

We thank the reviewer's attention to detail in this regard, and acknowledge that copying of survey items in quotations resulted in this formatting inconsistency in our original submission. 

Throughout the manuscript, we have now substituted single (‘’) and double (“”) curly/smart quotation marks in all instances where the original ones were straight. While the specific format for quotations is not explicitly stipulated as a PLoS style requirement (https://journals.plos.org/plosone/s/submission-guidelines#loc-style-and-format), curly quotes appear to be acceptable (e.g., https://doi.org/10.1371/journal.pone.0262562 and https://doi.org/10.1371/journal.pone.0263023).

2. Check references 

Uniform formatting should be followed for references. DOIs are missing for more references. Some references are having URLs. References can also be listed in alphabetical order.

In relation to published journal articles, PLoS submission guidelines stipulate that ‘DOI is acceptable … in addition to traditional volume and page numbers’ [but] ‘do not provide a … URL’ (https://journals.plos.org/plosone/s/submission-guidelines). In accordance with journal style requirements, our intention has been to include DOIs (where available) for all referenced journal articles. We have re-checked our reference list and discovered that the DOI for one journal reference (Zou 2004; reference #29) was missing in the original submission, and have added this in the revised version. 

To facilitate readers’ access to referenced material, we added URL hyperlinks—where available—to references for online material other than journal articles that lack DOIs. We believe this to in accordance with PLoS house style requirements. 

Further, also in accordance with house style, our bibliography presents references in ‘Vancouver’ style, i.e., sequentially numbered by the order in which each first appears in the text, rather than in alphabetical order by first author (i.e., ‘Harvard’ style).

---

## [Decision Letter · Decision Letter 1]

28 Mar 2023

Measuring for primary prevention: an online survey of local community perspectives on family and domestic violence in regional Australia

PONE-D-22-35475R1

Dear Dr. Woods,

We’re pleased to inform you that your manuscript has been judged scientifically suitable for publication and will be formally accepted for publication once it meets all outstanding technical requirements.

Kind regards,

Phyllis Lau, PhD

Academic Editor

PLOS ONE

Additional Editor Comments (optional):

Reviewers' comments:

Reviewer's Responses to Questions

**Comments to the Author**

1. If the authors have adequately addressed your comments raised in a previous round of review and you feel that this manuscript is now acceptable for publication, you may indicate that here to bypass the “Comments to the Author” section, enter your conflict of interest statement in the “Confidential to Editor” section, and submit your "Accept" recommendation.

Reviewer #2: All comments have been addressed

2. Is the manuscript technically sound, and do the data support the conclusions?

Reviewer #2: Yes

3. Has the statistical analysis been performed appropriately and rigorously? 

Reviewer #2: Yes

4. Have the authors made all data underlying the findings in their manuscript fully available?

Reviewer #2: Yes

5. Is the manuscript presented in an intelligible fashion and written in standard English?

Reviewer #2: Yes

6. Review Comments to the Author

Reviewer #2: The authors have updated the plots, quotes and references. They have done the necessary changes to the article as suggested. Now the paper can be accepted for publication.

7. PLOS authors have the option to publish the peer review history of their article (what does this mean?). If published, this will include your full peer review and any attached files.

Reviewer #2: No

---

## [Editor Report · Acceptance letter]

31 Mar 2023

PONE-D-22-35475R1 

Measuring for primary prevention: an online survey of local community perspectives on family and domestic violence in regional Australia 

Dear Dr. Woods:

I'm pleased to inform you that your manuscript has been deemed suitable for publication in PLOS ONE. Congratulations! Your manuscript is now with our production department. 

Kind regards, 

on behalf of

Dr. Phyllis Lau 

Academic Editor

PLOS ONE